# RefAny3D: 3D Asset-Referenced Diffusion Models for Image Generation

**Hanzhuo Huang[1]**    **Qingyang Bao[3]**    **Zekai Gu[4]**    **Zhongshuo Du[5]**
**Cheng Lin[6]**    **Yuan Liu[4†]**    **Sibei Yang[2†]**

[1]ShanghaiTech University,  [2]Sun Yat-sen University,  [3]University of Toronto,
[4]The Hong Kong University of Science and Technology,  [5]SynWorld,
[6]Macau University of Science and Technology

huanghanzhuo@shanghaitech.edu.cn, yuanly@ust.hk,
yangsb3@mail.sysu.edu.cn,

## Abstract

In this paper, we propose a 3D asset-referenced diffusion model for image generation, exploring how to integrate 3D assets into image diffusion models. Existing reference-based image generation methods leverage large-scale pretrained diffusion models and demonstrate strong capability in generating diverse images conditioned on a single reference image. However, these methods are limited to single-image references and cannot leverage 3D assets, constraining their practical versatility. To address this gap, we present a cross-domain diffusion model with dual-branch perception that leverages multi-view RGB images and point maps of 3D assets to jointly model their colors and canonical-space coordinates, achieving precise consistency between generated images and the 3D references. Our spatially aligned dual-branch generation architecture and domain-decoupled generation mechanism ensure the simultaneous generation of two spatially aligned but content-disentangled outputs, RGB images and point maps, linking 2D image attributes with 3D asset attributes. Experiments show that our approach effectively uses 3D assets as references to produce images consistent with the given assets, opening new possibilities for combining diffusion models with 3D content creation. Project page: https://judgementh.github.io/RefAny3D.

## 1 Introduction

In recent years, text-to-image diffusion models (Ho et al., 2020; Ramesh et al., 2022; Rombach et al., 2021; Saharia et al., 2022; Labs, 2024) have made remarkable progress in image synthesis, enabling the generation of high-quality and diverse images from textual prompts. However, relying solely on text prompts is often insufficient to capture fine-grained semantics and complex visual details (Witteveen & Andrews, 2022). This limitation is particularly pronounced in scenarios that require faithful preservation of a subject's identity, such as personalized content creation, advertising, marketing, and artistic design. Although existing methods (Gal et al., 2022; Ruiz et al., 2023; Ye et al., 2023; Li et al., 2023; 2024b; Shi et al., 2024; Cai et al., 2025; Tan et al., 2025) have concentrated on preserving object identity within 2D images, identity-preserving generation conditioned on 3D assets remains underexplored and represents a promising direction for future research.

Current subject-driven generation methods preserve identity by leveraging either local features or global semantics from reference images. Some methods (Gal et al., 2022; Hu et al., 2022; Ruiz et al., 2023) fine-tune diffusion models or text embeddings separately for each subject to capture fine-grained details, which is computationally expensive. More recent approaches (Shi et al., 2024; Cai et al., 2025; Tan et al., 2025) improve efficiency by introducing attention mechanisms between generated and reference images, enabling the model to effectively learn reference-guided generation. Other approaches (Ye et al., 2023; Li et al., 2023) map images into the text space to encode global semantics, which is computationally efficient but tends to oversimplify representations. Compress-

---

[†]Corresponding Authors.

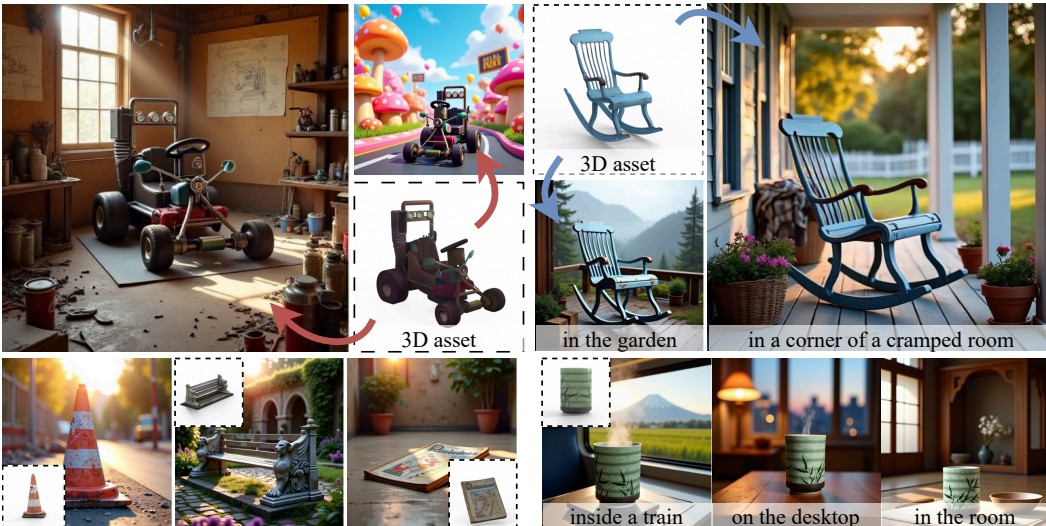

Figure 1: Results of our RefAny3D. Given a 3D asset, our method can generate high-quality and 3D asset-consistent images.

ing an image into a few text tokens results in the loss of spatial information, making it difficult for the generated images to faithfully correspond to the fine-grained details of the reference.

While extensive subject-driven generation (Tan et al., 2025; Kumari et al., 2025) has demonstrated that image generation conditioned on a single or a few 2D reference images can maintain identity consistency, 3D asset-referenced image generation remains unexplored. In practical applications, creators often require the direct use of 3D assets, such as meshes, as references to visualize how an object would manifest across diverse scenes and environments. Such scenarios go beyond the scope of purely 2D references and give rise to a new problem: how to leverage 3D assets as conditioning signals to generate images that are not only identity-consistent, but also geometry-consistent and texture-consistent.

Although existing methods (Cai et al., 2025; Tan et al., 2025) have achieved impressive results with 2D reference images, they remain fundamentally limited when extended to the task of generation conditioned on 3D assets. Unlike 2D settings, 3D asset-referenced generation requires not only semantic-level identity preservation, but also strict consistency with the geometry and texture of the reference 3D asset. This goes beyond the capability of existing methods. Specifically, first, the consistency capability of current methods remains inadequate for 3D asset-referenced generation, where the synthesized images are required to align precisely with the geometric structure and texture of the reference 3D asset. Second, approaches (Ye et al., 2023; Tan et al., 2025) limited to a single reference image are inherently unable to capture the full appearance of the object. Finally, in methods based on multi-image conditioning (Kumari et al., 2025; Zeng et al., 2024), or in straightforward extensions of existing approaches to multiple inputs, the absence of 3D structural priors prevents consistent spatial correspondence across views, leading to viewpoint conflicts and cross-view inconsistencies. In addition, recent image editing models (Labs et al., 2025; Wu et al., 2025) have shown strong instruction-following and image understanding capabilities. However, they remain insufficient for effectively addressing the challenge of 3D alimitationsset-referenced generation. A straightforward way to leverage such models is to manually select a viewpoint of the 3D asset, render it into an image, and then apply editing instructions to the rendered view. The primary shortcoming of this approach is that the resulting images often suffer from foreground–background inconsistency and may hallucinate non-existent content. Overall, the key challenge of 3D asset-referenced generation lies in effectively leveraging the structural and textural priors of 3D assets to achieve faithful, view-consistent, and detail-preserving image synthesis.

In this paper, we propose RefAny3D , a 3D asset-referenced and 3D structure-aware image generation framework, which is designed to synthesize images with faithful fidelity and consistent alignment to the 3D assets. The core idea is to construct a 3D-aware generative framework that leverages the correspondence between normalized object coordinates (point maps) (Wang et al., 2019) and their associated RGB values, thereby ensuring consistent alignment with the 3D assets. This consis-

tency stems from two key properties of point maps. First, while texture information alone may introduce ambiguities or repetitions across different views, point maps are uniquely tied to the object's geometry, enabling more reliable cross-view correspondence. Second, point maps are continuous and invariant to object pose or position, making them easier to learn and more effective anchors for linking geometric structure with texture. Specifically, we formalize the generation process as modeling the joint distribution of RGB appearance and point maps. Conditioned on multi-view RGB images and point maps of the 3D asset, the framework is trained to simultaneously generate photorealistic images of the object together with their corresponding point maps. To achieve this, we introduce a spatially aligned, domain-decoupled dual-branch generation strategy that enables the model to synthesize both RGB images and point maps in a unified manner. Unlike prior approaches, our method explicitly leverages object coordinates to build structural awareness of the 3D object, while the point maps establish pixel-level correspondences across different views, which is otherwise difficult to achieve without such guidance. Consequently, our approach maintains faithful consistency of complex geometry and texture.

In summary, our main contributions are as follows: (1) We propose a 3D asset-referenced image generation framework that ensures faithful alignment and consistency with the underlying 3D assets. (2) We design a spatially aligned, domain-decoupled dual-branch generation strategy that enables the model to jointly generate RGB images and point maps, thereby enhancing its 3D structural awareness. (3) We demonstrate that our approach achieves accurate preservation of the visual identity of 3D objects. Extensive qualitative and quantitative evaluations show that it consistently outperforms existing baselines on the 3D asset-referenced generation task, delivering superior fine-grained consistency and robust fidelity even for complex models with intricate geometric and textural details.

## 2 RELATED WORK

**Diffusion Models.** Recently, diffusion models (Ho et al., 2020; Nichol & Dhariwal, 2021; Ramesh et al., 2022; Rombach et al., 2021) trained on large-scale datasets (Schuhmann et al., 2022; Byeon et al., 2022) have achieved significant breakthroughs in generating photorealistic and diverse visual content, excelling across a wide range of image generation tasks (Huang et al., 2023; 2025; Gu et al., 2025), including image editing (Batifol et al., 2025; Wu et al., 2025), controllable content generation (Zhang et al., 2023), and subject-driven generation (Gal et al., 2022; Ruiz et al., 2023; Ye et al., 2023). Pioneering works (Ramesh et al., 2022; Rombach et al., 2021) first showcased the strong generative and generalization capabilities of diffusion models trained on large-scale datasets. To further enhance their generative capability, Transformer (Vaswani et al., 2017; Zhu et al., 2025; Dai et al., 2025; Dai & Yang, 2025; Wei et al., 2025; Tang et al., 2025a; Yang et al., 2025; Zheng et al., 2025; Tang et al., 2025b; Qian et al., 2025; Yang et al.; Zhang et al., 2025; Shi et al., 2025) architectures have been incorporated into diffusion models (Peebles & Xie, 2023), enabling greater scalability. More recent model (Labs, 2024) adopts flow-matching (Lipman et al., 2022) training in conjunction with MMDiT (Esser et al., 2024) architectures and large-scale datasets, achieving state-of-the-art performance in text-to-image generation. Despite these advances, text-to-image models still lack effective approaches for generating images conditioned on 3D assets as references.

**Subject-Driven Generation.** The goal of subject-driven generation is to capture the characteristics of a given reference subject, enabling the synthesis of realistic images of the subject across diverse scenes. Early methods (Gal et al., 2022; Hu et al., 2022; Ruiz et al., 2023) typically adapt the text embedding layer (Gal et al., 2022) or the model (Hu et al., 2022; Ruiz et al., 2023) using only a few reference images, while applying regularization (Ruiz et al., 2023) to maintain the model's generalization capability. Because these methods require retraining and fine-tuning for each subject, they entail significant computational and time costs, limiting their practical applicability. To reduce training overhead, some works (Ye et al., 2023; Li et al., 2023) learn an adapter from image space to text space, enabling direct encoding of images into text embeddings without per-subject training. However, these approaches compress an image into only a few text tokens, often limiting fine-grained fidelity to the reference. Recent approaches (Cai et al., 2025; Tan et al., 2025) concatenate the generated and reference images into a unified token sequence and leverage shared attention to better capture fine-grained correspondences, addressing the challenge of limited detail fidelity. Beyond image-guided subject-driven generation, several works (Wu & Zheng, 2022; Wu et al., 2023; Wang et al., 2024a;b) investigate 3D-guided generation using single or few 3D exemplars. Wu & Zheng (2022) generate 3D shapes from a single reference 3D shape using multi-scale 3D representations;

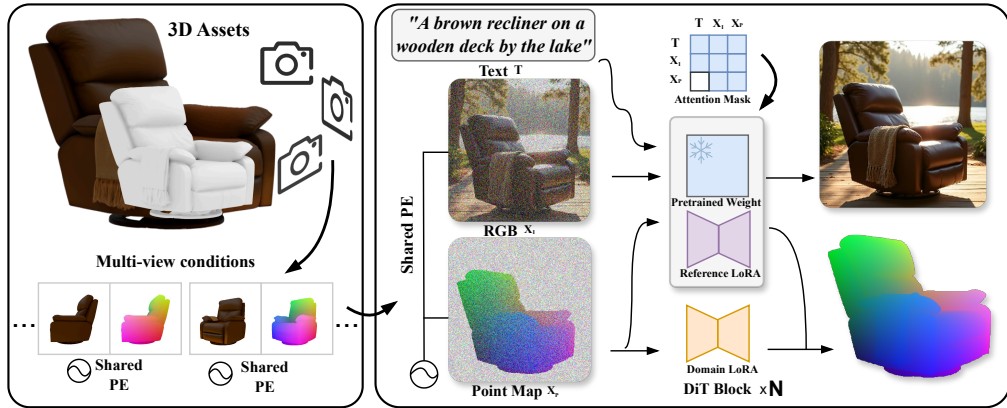

Figure 2: Overview of RefAny3D. Given a 3D asset, we render multi-view inputs as conditioning signals for the diffusion model and simultaneously generate the point map of the target RGB image. To ensure pixel-level consistency across different viewpoints, we adopt a shared positional encoding strategy. Moreover, to disentangle the RGB domain from the point map domain, we incorporate Domain-specific LoRA and Text-agnostic Attention. Benefiting from this 3D-aware disentanglement design, our method is able to generate high-quality images that maintain strong consistency with the underlying 3D assets.

Sin3DM (Wu et al., 2023) learns a diffusion model from a single textured 3D shape; ThemeStation (Wang et al., 2024a) produces theme-aware 3D assets from few exemplars; and Phidias (Wang et al., 2024b) enables text-, image-, and 3D-conditioned content creation via reference-augmented diffusion. While these methods leverage 3D inputs, they primarily focus on 3D asset generation, rather than using provided 3D assets to guide 2D subject-driven image generation. Despite their merits, these methods often lack precision when using 3D assets with complex textures as references. In contrast, our approach effectively leverages 3D structural cues to achieve more faithful and consistent reference generation.

**Multi-modal Image Generation.** Recent studies (Ke et al., 2024; Long et al., 2024; Yang et al., 2024; Li et al., 2024a; Huang et al., 2024; Fu et al., 2024; He et al., 2024; Ye et al., 2024; Zhang et al., 2024) have demonstrated that diffusion models are capable of generating not only high-fidelity RGB images but also diverse physical property images, such as albedo, normal, roughness, and irradiance. Several recent multi-view generation works (Long et al., 2024; Li et al., 2024a; Huang et al., 2024) improve 3D reconstruction quality by jointly generating multi-view normals and color images. Moreover, an increasing number of studies (Ke et al., 2024; He et al., 2024; Fu et al., 2024; Ye et al., 2024) leverage the multi-modal generation capability of diffusion models to perform dense prediction tasks. These methods typically condition on RGB images to predict pixel-aligned normal or depth maps. However, these multi-modal image generation methods either jointly generate multiple modalities but remain constrained to multi-view settings (Liu et al., 2023), or they simply translate one modality into another single modality (Ke et al., 2024; Gu et al., 2025). In contrast, our approach simultaneously generates information across multiple modalities and is not restricted to object-centric multi-view settings.

## 3 METHOD

We propose a 3D asset-conditioned image generation framework, which jointly models RGB image and point map distributions to create high-quality images for a 3D object. The overall generation pipeline is shown in Fig. 2.

**Overview.** Given a conditional 3D object $y$, we represent it as a set of multi-view RGB–point map pairs $\{(C_{I_i}, C_{P_i})|i = 0, 1, \cdots, N\}$, where $N$ is the total number of views, $C_{I_i} \in \mathbb{R}^{h \times w \times 3}$ denotes the RGB images from the $i$-th viewpoint, and $C_{P_i} \in \mathbb{R}^{h \times w \times 3}$ denotes the corresponding rasterized 3D coordinates of the object. The RGB images and point maps are pixel-wise aligned,

jointly encoding the color and its associated 3D position. Formally, our objective is to learn the conditional distribution given a reference 3D model $y$ and a text prompt $c$

$$p(x_I, x_P | y, c)$$

where $x_I$ denotes the target RGB image, and $x_P$ denotes its corresponding point map. To accurately preserve the visual identity of a 3D object, we adopt a dual-branch conditional generation framework that jointly generate the RGB image and point map, leveraging 3D spatial constraints to reinforce reference consistency (3.1). However, generating spatially aligned images from two domains introduces the challenge of texture bleeding. To address this, we propose a domain decoupling generation strategy (3.2). For training RefAny3D , we prepare an object pose-aligned datasets, which consists of images containing the objects of interest, their corresponding 3D object models, and the associated object poses (3.3).

## 3.1 Spatially Aligned Dual-Branch Generation

We simultaneously generate spatially aligned RGB images and point maps to provide precise 3D spatial information of the reference object for conditional image generation. To achieve conditional generation for the diffusion model, similar to prior works (Tan et al., 2025; Wang et al., 2025), we concatenate the target tokens and condition tokens into a unified sequence. Furthermore, to generate spatially-aligned cross-domain images, we employ shared positional embeddings.

**Conditional Token Sequence.** We encode the RGB images and point maps of the 3D model into latent conditional tokens using a pretrained VAE encoder, which are subsequently concatenated with the noisy target latent. To preserve the fidelity of the conditional features, we set the timestep of the conditional tokens to $0$ during the diffusion denoising process.

**Shared Positional Embedding for Cross-Domain.** To maintain spatial alignment between the RGB image and point map during generation, we apply shared positional encodings to tokens across both domains. This approach exploits an inherent property of DiT, induced by positional encoding, to naturally assign higher attention scores to tokens with the same positional embeddings. To mitigate biases caused by inconsistent distances among conditional tokens, we introduce a unified positional shift term. Specifically, for a conditional token at spatial position $(i, j)$, the positional encoding is set to $(i - w, j)$, where $w$ is the width of the target latent image. This shift guarantees that the conditional tokens and target tokens remain spatially disjoint.

## 3.2 Domain Decoupling Generation

The core challenge of jointly generating point maps and RGB images lies in their inherent information asymmetry. A point map defines only the object's 3D geometry and pose, while the RGB image contains photorealistic details of the entire scene. In a unified framework, this asymmetry often causes the point map, which lacks background information, to be affected by interference from the RGB branch and text prompt. Therefore, we introduce domain-specific LoRA and a text-agnostic attention to achieve accurate generation in both domains.

**Domain-specific LoRA.** We decouple domain knowledge using a domain switcher and a dual LoRA structure. The domain switcher specifies the domain of each token to guide the generation of RGB images and point maps. Specifically, we associate each domain with a learnable embedding, which is then concatenated with the timestep embedding. To further decouple the learning of domain-specific knowledge, we also introduce two independent LoRA (Hu et al., 2022) modules, termed Reference-LoRA and Domain-LoRA, to separately learn the 3D object reference generation and point map domain generation. The Reference-LoRA is activated for all conditioning tokens to learn general appearance features, while the specialized Domain-LoRA is activated only for point map tokens to learn specific geometric information. This design enables the model to generate high-fidelity point maps and RGB images.

**Text-agnostic Attention.** To further suppress background information leakage into the point map, we introduce a text-agnostic attention mask in the point map branch. This design minimizes the influence of text tokens on the point map, as textual descriptions often contain substantial background information that is irrelevant to the point map. In contrast, the RGB tokens can attend to all tokens, allowing them to fully exploit the information from both the text and the point map. This design

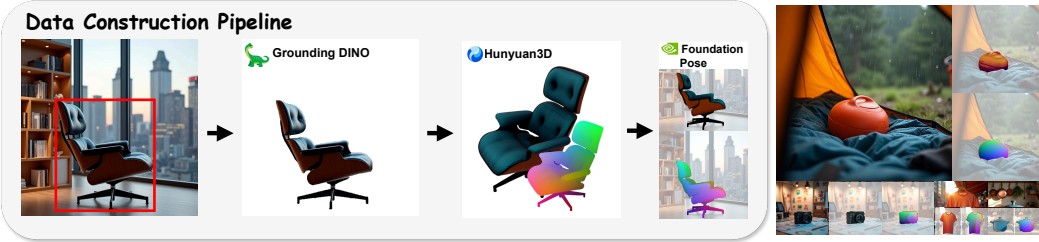

(a) Data construction pipeline         (b) Examples from the dataset

Figure 3: (a) Data construction pipeline. We first use GroundingDINO (Liu et al., 2024) to extract the objects of interest, then convert the images into 3D models using Hunyuan3D (Zhao et al., 2025), and finally apply FoundationPose (Wen et al., 2024) to estimate the poses of the 3D models in the images. (b) Examples from the dataset.

ensures the point map is generated as a geometric proxy, reducing the influence of background corruption, while the RGB branch can fully utilize the accurate geometric guidance to render detailed shapes and textures.

## 3.3 DATASETS

To train our model for 3D asset-reference generation, we require an object pose-aligned dataset. Specifically, this dataset is composed of images containing the objects of interest, their corresponding 3D assets, and the associated object poses, which are used to generate the corresponding point maps. However, existing public datasets do not provide all the required data. We build upon Subjects200k, a subject-driven generation image dataset, and further enhance it by incorporating 3D assets along with object pose annotations for each image. The overall construction pipeline is illustrated in Fig. 3. First, for each image, we use the object names provided in Subjects200k as prompts to GroundingDINO (Liu et al., 2024) to extract the corresponding objects of interest. Next, we convert each extracted object from the image into a 3D asset using Hunyuan3D (Zhao et al., 2025). Finally, taking the generated 3D asset as input, we estimate its pose using FoundationPose (Wen et al., 2024).

## 4 EXPERIMENT

### 4.1 EXPERIMENTAL SETTINGS

**Implementation details.** We use Flux.1-dev as our base model. Following Tan et al. (2025), we train the model with the Prodigy optimizer (Li et al., 2023). Our model is trained for 30k steps on 8 H800 GPUs. To enable classifier-free guidance (Ho & Salimans, 2022), we randomly drop the text and the reference multi-view images with a probability of 0.1 each. The number of views in the multi-view images is set to 8.

**Baselines.** We adopt Textual Inversion (Gal et al., 2022), DreamBooth (Ruiz et al., 2023), IP-Adapter (Ye et al., 2023), DSD (Cai et al., 2025), and OminiControl (Tan et al., 2025) as our baseline methods. For personalized text-to-image generation methods that require training (e.g., Textual Inversion and DreamBooth), we use multi-view images of each 3D asset as training data, fine-tune the model accordingly, and then sample from the customized model to obtain generated images. For methods that do not require additional training (e.g., IP-Adapter, DSD, and OminiControl), we use their official pre-trained models. Since these models do not support multiple images as input conditions, we select a single view of each 3D asset as input to generate images for comparison with our approach.

**Metrics.** We employ both foundational vision models and LVLMs to comprehensively evaluate ID consistency, texture consistency, and aesthetic quality of the generated images and 3D assets. Specifically, we measure semantic consistency by computing CLIP (Radford et al., 2021) and DINO (Caron et al., 2021) feature similarities between the generated images and the multi-view renderings of the 3D assets. In addition, we compute the CLIP text–image similarity score for

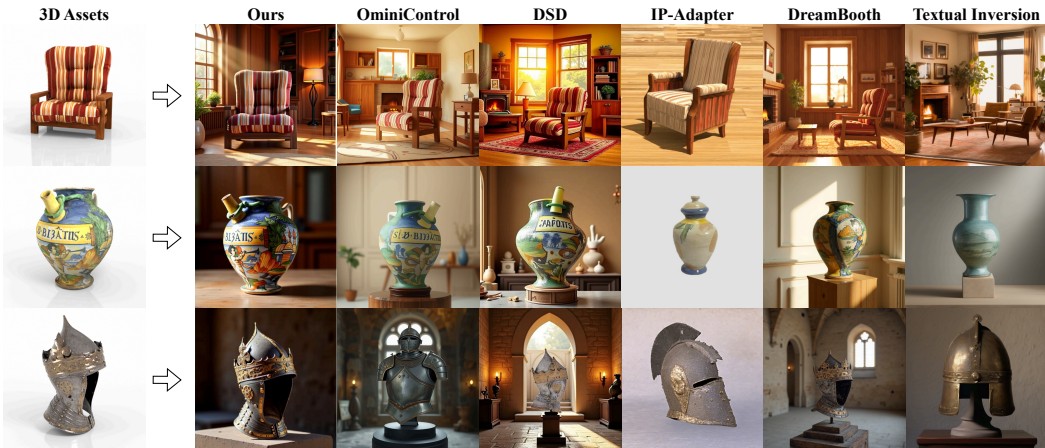

Figure 4: Qualitative comparison with other methods. Our approach achieves superior geometric and texture consistency compared to alternative methods.

| Method | GPT-eval | | | | CLIP | | | DINO | | GIM |
|---|---|---|---|---|---|---|---|---|---|---|
| | Texture↑ | Geometric↑ | Aesthetic↑ | Overall↑ | Img/Avg.↑ | Img/Max.↑ | Text↑ | Avg.↑ | Max.↑ | Count↑ |
| Textual Inversion | 2.894 | 4.421 | 6.263 | 4.526 | 0.827 | 0.878 | 0.323 | 0.548 | 0.653 | 3359.895 |
| DreamBooth | 5.368 | 6.684 | 6.894 | 6.315 | 0.867 | 0.912 | 0.328 | 0.695 | 0.809 | 3483.368 |
| IP-Adapter | 3.833 | 5.278 | 5.167 | 4.759 | 0.863 | 0.913 | 0.312 | 0.652 | 0.760 | 3137.167 |
| DSD | 4.842 | 6.473 | 7.105 | 6.140 | 0.832 | 0.884 | 0.329 | 0.644 | 0.761 | 3568.737 |
| OminiControl | 5.631 | 6.578 | 6.893 | 6.367 | 0.855 | 0.901 | 0.332 | 0.665 | 0.783 | 3474.211 |
| **Ours** | **6.315** | **7.368** | **7.687** | **7.123** | **0.873** | **0.923** | **0.340** | **0.720** | **0.843** | **3901.316** |

Table 1: Quantitative results comparing our method against the baselines, with the **best** scores highlighted in bold and the second-best underlined. Our approach achieves the best performance across all evaluation metrics, including both GPT-based measures and baseline model metrics. In particular, our method yields substantial gains on the GPT-eval Texture and Geometric scores, as well as the GIM metric, which are particularly indicative of fine-grained geometric and textural fidelity. These results demonstrate the effectiveness of our framework in faithfully preserving 3D asset consistency and generating high-quality outputs beyond existing baselines.

each generated image and its corresponding text prompt. We assess texture consistency by using GIM (Shen et al., 2024) to count the number of matched keypoints between the generated images and the multi-view images. Furthermore, we leverage GPT evaluation by providing both the generated and multi-view images to GPT-5 to obtain scores on textual consistency, geometric consistency, and aesthetic quality, thereby enabling a more comprehensive assessment of the generation results. We then compute the average of these three scores as an overall score.

## 4.2 COMPARISONS

**Qualitative results.** As shown in Fig. 4, we present qualitative comparisons between our method and baseline methods. Our method achieves superior geometric and texture consistency with the 3D assets compared to others. In the first row of Fig. 4, our method accurately captures the undulating geometric surface of the chair cushion, whereas other methods fail to reproduce these fine structures. For 3D assets with complex textures (second row), our method faithfully reproduces the characters and illustrations on the vase, while other methods fail to maintain such consistency. Moreover, unlike DreamBooth (Ruiz et al., 2023) and Textual Inversion (Gal et al., 2022), our method requires no additional training on the reference 3D asset to produce consistent results, highlighting its practical advantage. Fig. 5 further shows the point map results generated by our method, demonstrating that it can simultaneously generate the foreground object and its corresponding point map to indicate relative coordinate relationships. In addition, our method can be integrated with multi-view to 3D generation models (Zhao et al., 2025), enabling the synthesis of images conditioned on multiple reference views. Fig. 6 illustrates the effectiveness of our approach on multi-view inputs. By modifying the prompt or adjusting the 3D asset's local coordinate system, the generated images exhibit diverse perspectives, as illustrated in Fig. 9.

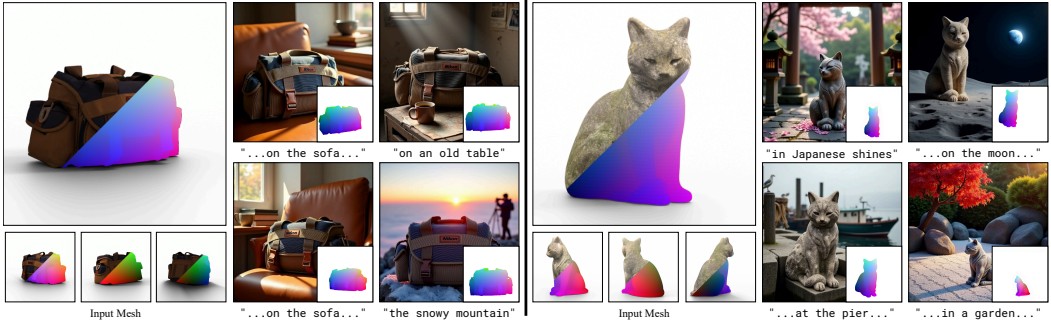

Figure 5: Qualitative results with different 3D assets as references. Our method takes a given 3D mesh as input and generates both RGB images and point maps in a unified manner. By enforcing pixel-level spatial alignment between the point maps and RGB outputs, the framework ensures consistent geometry–texture correspondence across views. Moreover, the incorporation of point maps enhances the model's 3D structural awareness, thereby improving the fidelity and consistency of image generation with respect to the reference 3D assets.

| Method | Faithful↑ | ID↑ | Aesthetic↑ | Rank↓ |
|---|---|---|---|---|
| Textual Inversion | 2.182 | 3.053 | 3.526 | 5.158 |
| DreamBooth | 3.836 | 4.315 | 4.421 | 2.526 |
| IP-Adapter | 1.982 | 2.947 | 2.053 | 5.737 |
| DSD | 4.145 | 4.368 | 4.158 | 2.842 |
| OminiControl | 3.909 | 4.263 | 4.526 | 3.158 |
| **Ours** | **4.655** | **4.737** | **4.632** | **1.579** |

Table 2: Quantitative results of the user study. We evaluate 3D consistency (Faithful), identity preservation (ID), aesthetic quality, and overall ranking (Rank).

Figure 6: Qualitative results on multi-view images. Our method can be integrated into existing multi-view image-to-3D generation pipelines.

**Quantitative Results.** The quantitative evaluation results comparing our method with other baselines are shown in Table 1. We report the evaluation results from both large-scale vision-language models (LVLMs) and vision foundation models. For LVLM evaluation, we employ GPT-5 to assess texture consistency, geometric consistency, and aesthetic quality, and additionally report an overall score as their average. Specifically, we concatenate the generated image and the multi-view images of the 3D asset into a 3×3 grid and prompt GPT-5 to rate the generated image on each metric from 0 to 10, where higher scores indicate better consistency and quality. Our method achieves the best performance across all GPT metrics compared to other baselines, with notably superior results on texture and geometric consistency. For evaluation using vision foundation models, we adopt CLIP (Radford et al., 2021) and DINO (Caron et al., 2021) as image encoders to compute feature similarities as a measure of semantic consistency. Since the reference consists of multiple images, we compute both the average and the maximum similarity between the generated image and the multi-view references to obtain a more comprehensive assessment. To reduce background effects on the CLIP scores, we remove the background from all images before comparing them with the reference renderings. Our method outperforms all other approaches on CLIP and DINO metrics except IP-Adapter (Ye et al., 2023).

To further capture fine-grained correspondences, we employ GIM (Shen et al., 2024), a state-of-the-art image matching method, to compute the number of matched keypoints between the generated image and the multi-view references, thereby quantifying the correspondence with the real 3D asset details. Our method also outperfor ms all competing baselines on the vision foundation model metrics. As shown in Table 2, we also conduct a user study to evaluate generation quality along four dimensions: Faithfulness, Identity, Aesthetic Quality, and Overall Rank. Faithfulness measures the 3D consistency between the generated image and the reference 3D asset, while Identity evaluates whether the generated object preserves the identity of the 3D asset. Aesthetic Quality reflects the visual appeal of the generated image, and Overall Rank asks participants to provide a holistic com-

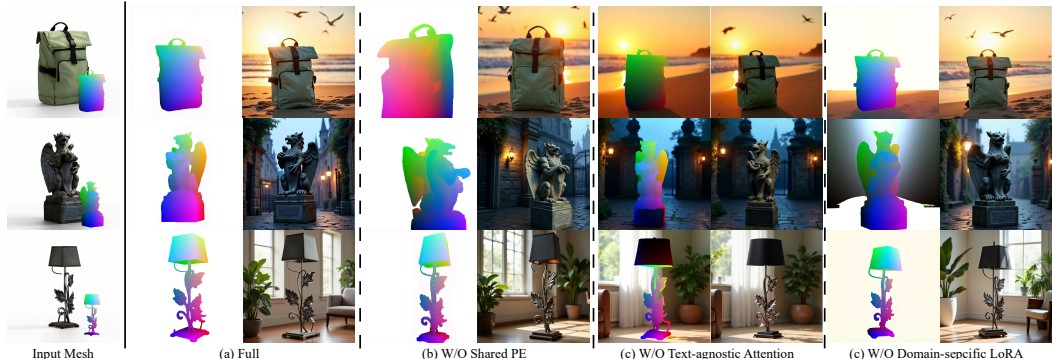

Input Mesh      (a) Full      (b) W/O Shared PE      (c) W/O Text-agnostic Attention      (c) W/O Domain-sepcific LoRA

Figure 7: Ablation studies on different components of our method: (a) full model; (b) without Shared Positional Embedding for Cross-Domain; (c) without Text-agnostic Attention; (d) without Domain-specific LoRA.

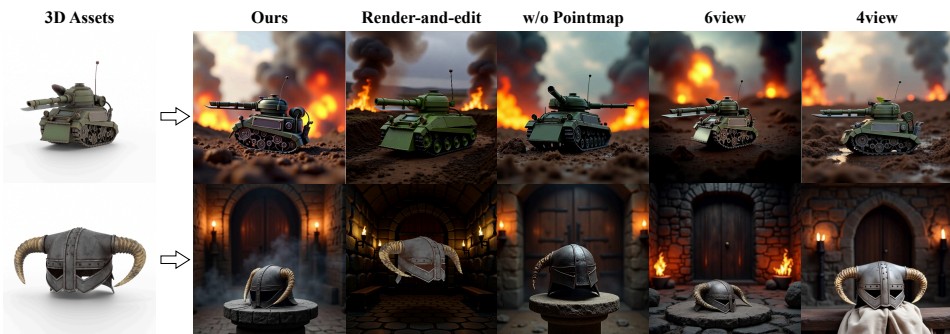

3D Assets      Ours      Render-and-edit      w/o Pointmap      6view      4view

Figure 8: Comparisons of ablation studies and the editing-based baseline.

parison across methods. The results indicate that our method is competitive across all metrics, with particularly strong performance in Faithfulness and Identity consistency.

## 4.3 DISCUSSIONS

In this section, we further design a set of experiments to validate the effectiveness of our proposed components.

**Without Shared Positional Embedding for Cross-Domain.** As shown in Fig. 7 (b), we remove the specially designed positional embeddings and train the model with token sequences arranged in their natural order, then compare the results against our full model. This comparison demonstrates the necessity of sharing positional embeddings between the RGB and point map under the same viewpoint. Without shared positional embeddings, the network lacks positional priors and struggles to learn accurate pixel-level correspondences between the point maps and RGB images. The resulting misalignment leads to degraded geometric consistency with the reference 3D asset. For example, the top of the "backpack" and the overall contour of the "griffin" fail to remain consistent with the reference.

**Without Domain-Decoupling Generation.** To evaluate the effectiveness of our proposed cross-domain decoupling generation strategy, we trained two models without the Domain-specific LoRA and Text-agnostic Attention modules and tested their performance. As shown in Fig. 7 (b) and (c), the generated point maps and RGB images exhibit color blending, particularly in the background regions of the point maps. When Text-agnostic Attention is removed, the point maps are influenced by the input text, which often contains rich background semantics, causing the point map background to align with that of the RGB branch. In addition, without Domain-specific LoRA, although the influence of text semantics is mitigated, a single LoRA is overburdened with simultaneously generating two domains and learning reference consistency, which still results in background artifacts and further degrades the overall generation quality.

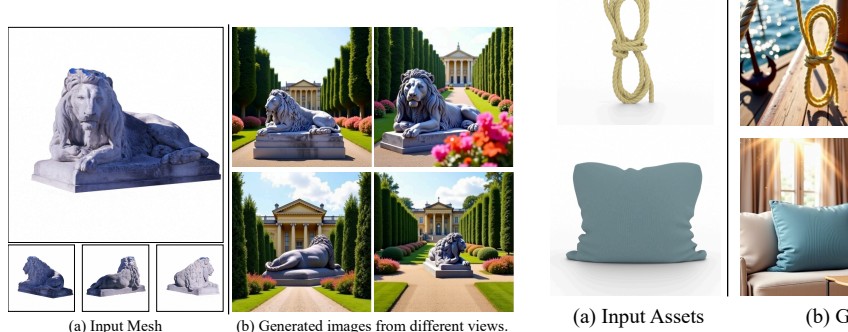

(a) Input Mesh (b) Generated images from different views.  (a) Input Assets (b) Generated Images

Figure 9: An example of controllable generation of object images from different viewpoints.

Figure 10: Limitation on non-rigid objects. While our method achieves high fidelity to the input 3D assets, it does not account for physical interactions in the scene.

**Without Pointmap Generation.** To assess the role of pointmap prediction, we trained a model without the pointmap. As shown in Fig. 8, removing this branch eliminates explicit 3D cues, making training unstable and resulting in poor 3D consistency and mismatch with the reference asset.

**Number of Conditional Views.** To evaluate the effect of the number of views on the results, we trained models with 6 and 4 views and compared them with our 8 view model. As shown in Fig. 8, the results show that our method still works effectively with fewer views, and performance consistently improves as the number of views increases.

**Comparison with Editing-based Methods.** A straightforward approach to generating images of 3D objects is to first render the 3D model and then apply an editing model to modify the rendered image. However, this requires manual selection of viewpoints, object placement, and renderer quality. Poor choices can lead to foreground-background mismatches or unrealistic floating objects. Furthermore, the method is limited to a single visible viewpoint. For novel viewpoints, the editing model may hallucinate nonexistent parts, leading to inconsistencies with the 3D model. We tested using Qwen-Image-Edit-2509 (Wu et al., 2025), and the results are shown in Fig. 8. For the rear of the tank, the model hallucinates parts that are inconsistent with the 3D asset. For the helmet, the editing model clearly produces a mismatch between the rendered foreground and the real background, with the object unrealistically floating in midair.

## 5 CONCLUSION

In this paper, We propose RefAny3D a new 3D asset-referenced image generation framework that possesses 3D awareness and can synthesize high-quality images with precise consistency to the given references. Our key idea is to leverage normalized object coordinates (point maps) as structural anchors, jointly modeling RGB appearance and point maps to achieve reliable geometry–texture correspondence. To this end, we design a spatially aligned, domain-decoupled dual-branch strategy that enables simultaneous generation of RGB images and point maps, thereby enhancing the model's structural awareness. Experiments demonstrate the effectiveness of our approach, showing that it delivers superior fine-grained consistency and robust fidelity compared to existing baselines on the 3D asset-referenced generation task. **Limitations.** Although RefAny3D enables 3D asset-referenced generation with strong geometric and texture consistency, it is less effective in handling non-rigid object references due to dataset limitations. As demonstrated in Fig. 10, deformable assets such as ropes and cushions retain an rigidity. In addition, conditioning the diffusion model on an extended number of viewpoints introduces significant computational and time overhead. Nevertheless, this limitation could be alleviated in the future by employing more efficient attention optimization strategies to improve computational efficiency.

**Acknowledgments:** This work is supported in part by the National Natural Science Foundation of China under Grant No.62576365.

ETHICS STATEMENT

This work focuses on developing and evaluating image generation models. We acknowledge that such models carry potential ethical risks, particularly in relation to the generation of synthetic images that could be misused for the creation of deceptive or misleading content. In addition, the outputs of generative models may inadvertently infringe upon copyright or intellectual property rights if they resemble existing works too closely. To mitigate these risks, our experiments were conducted solely for research purposes, and all generated examples in this paper are used exclusively for scientific illustration. We emphasize that our work does not intend to promote harmful applications, and further safeguards, such as watermarking, usage policies, and responsible release practices, should be considered in future deployment of such systems.

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

# A APPENDIX

## A.1 DATASETS DETAILS

To train our 3D asset-reference generation model, we require a dataset of pose-aligned objects. We construct this dataset using Subjects200k, a large-scale subject-driven image collection. Our process begins by filtering for high-quality images, selecting those with the highest image quality scores provided in the dataset. Next, we remove the backgrounds from these images to prepare them for mesh extraction with the HunYuan 3D model (Zhao et al., 2025). For each image, we leverage the object names supplied by Subjects200k as prompts for GroundingDINO (Liu et al., 2024), which generates bounding boxes to accurately localize the objects of interest. Then we use Segment Anything Model (SAM) (Kirillov et al., 2023) to obtain the objects within the bounding box. The foreground generated by the SAM model may not be very good, so we set a threshold for the mask area ratio, and discard the results that exceed it. With the obtained foregrounds, we place all these objects into Hunyuan3D to first generate a white model. Since Hunyuan3D-Paint is too slow in unfolding UVs, we first reduce the number of faces of the objects and then use Open3D's UV-unwrapping script to handle this task, which significantly improves the generation speed. For pose estimation, the inputs consist of RGB images of the target object, depth maps estimated from these images using Depth Pro, object masks that segment the target from the background, and a reference 3D mesh model. The masked depth maps are calibrated to match the canonical scale of the mesh, ensuring consistency between the observed geometry and the model. With the calibrated depth and object mask, FoundationPose is employed to estimate the object's 6D pose relative to the camera. From the aligned model, a scene pointmap is rendered, where each pixel encodes the 3D coordinates of the visible surface point in the camera frame. The outputs include the estimated 6D pose, the rendered pointmaps (both standalone and overlaid on the input image), and consolidated RGB-D samples with pose annotations. This filtering method excludes failure cases such as incorrect orientations, mismatched viewpoints, or object localization errors. To further ensure that the generated 3D mesh truly matches the appearance of the object in the 2D image, we place the 3D asset at the estimated pose and compute the LPIPS between the rendered object and the corresponding image region, retaining only samples with an LPIPS value below 0.3. This step filters out texture mismatches or reconstruction artifacts in the 3D assets.These two complementary checks, Mask IoU for geometric and pose alignment and LPIPS for texture and appearance fidelity, form a reliable data filtering pipeline. After this filtering step, we use the estimated pose together with the 3D mesh, we additionally rendered videos showing the object rotating around its axis as well as the corresponding rotating pointmaps. Last, the remaining valid samples were consolidated and organized into the training dataset.

## A.2 IMPLEMENTATION DETAILS

The training is conducted at a resolution of $512 \times 512$, with the LoRA rank fixed to 16. Multi-view conditioning is realized by uniformly sampling $N = 8$ viewpoints at equal angular intervals, ensuring balanced geometric coverage. The model is trained on 8 NVIDIA H800 GPUs for 30k steps (approximately eight days), starting from the Flux-dev pretrained checkpoint. To incorporate multi-view conditions, we adopt the MMDiT architecture, which enables effective fusion of multimodal signals. To address the asymmetry between point maps and RGB images, we introduce Domain-specific LoRA and Text-agnostic Attention. Specifically, domain knowledge is decoupled via a domain switcher and dual-LoRA structure: a Reference-LoRA learns general appearance features across all tokens, while a Domain-LoRA is activated only for point map tokens to capture geometric information. In parallel, a text-agnostic attention mask suppresses the influence of background information from text tokens on the point map branch, ensuring that point maps serve as purely geometric proxies, while RGB tokens can fully exploit both semantic and geometric cues. Our framework thus comprises two coordinated branches, one generating geometry-oriented point maps and the other producing photorealistic RGB images, achieving consistent and high-fidelity results across both domains.

## A.3 THE USE OF LARGE LANGUAGE MODELS

In the process of preparing this paper, we employed large language models (LLMs) to polish the writing. Specifically, LLMs were used to improve the clarity, fluency, and coherence of our expressions without altering the substantive content or arguments. All core ideas, analyses, and conclusions

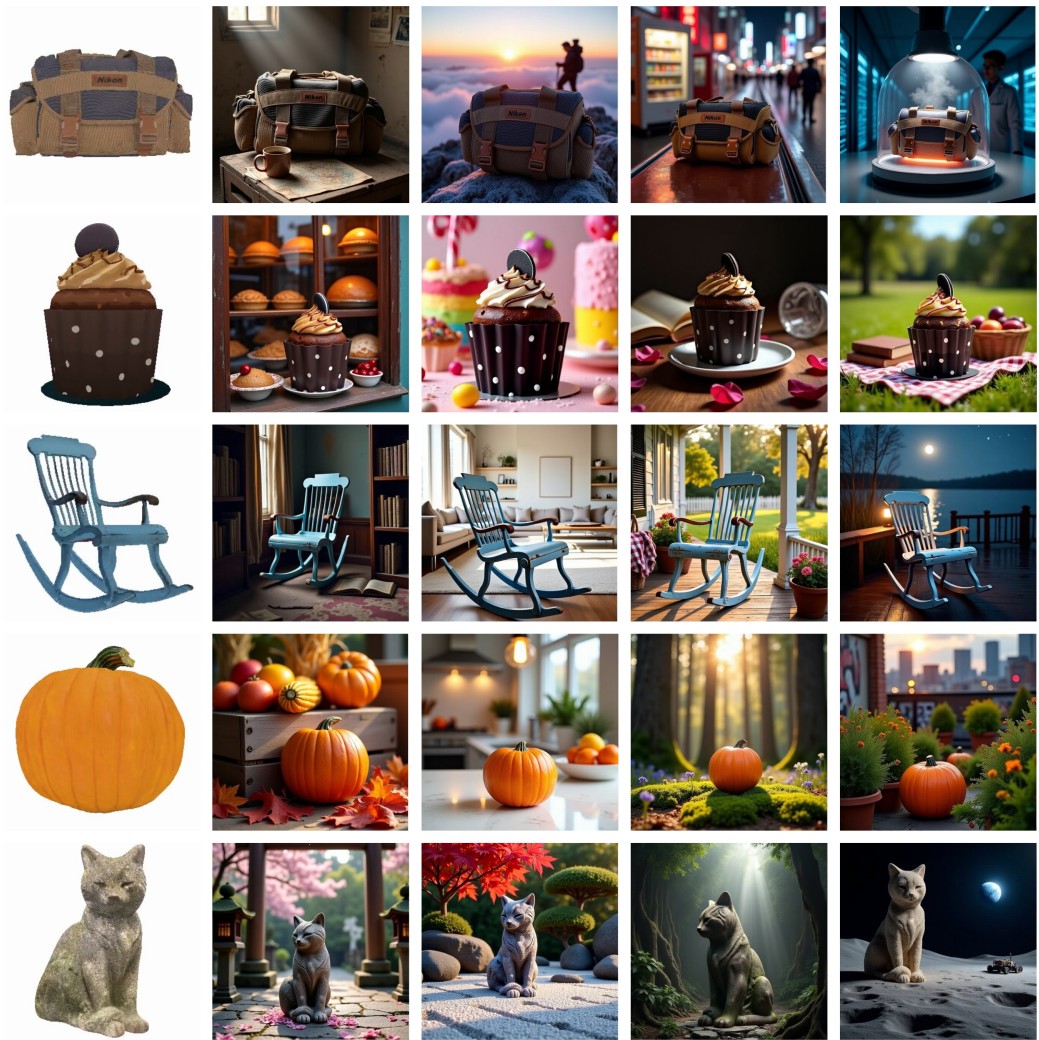

Figure 11: Qualitative results with different 3D assets as references.

were developed independently by the authors, while the LLM served solely as a language refinement tool to ensure readability and academic style.

### A.4 SUPPLEMENTARY QUALITATIVE RESULTS

We present additional experimental results using 3D assets as references, as shown in Figures 11 and 12. These examples further demonstrate the consistency with the 3D references and the high quality of our results.

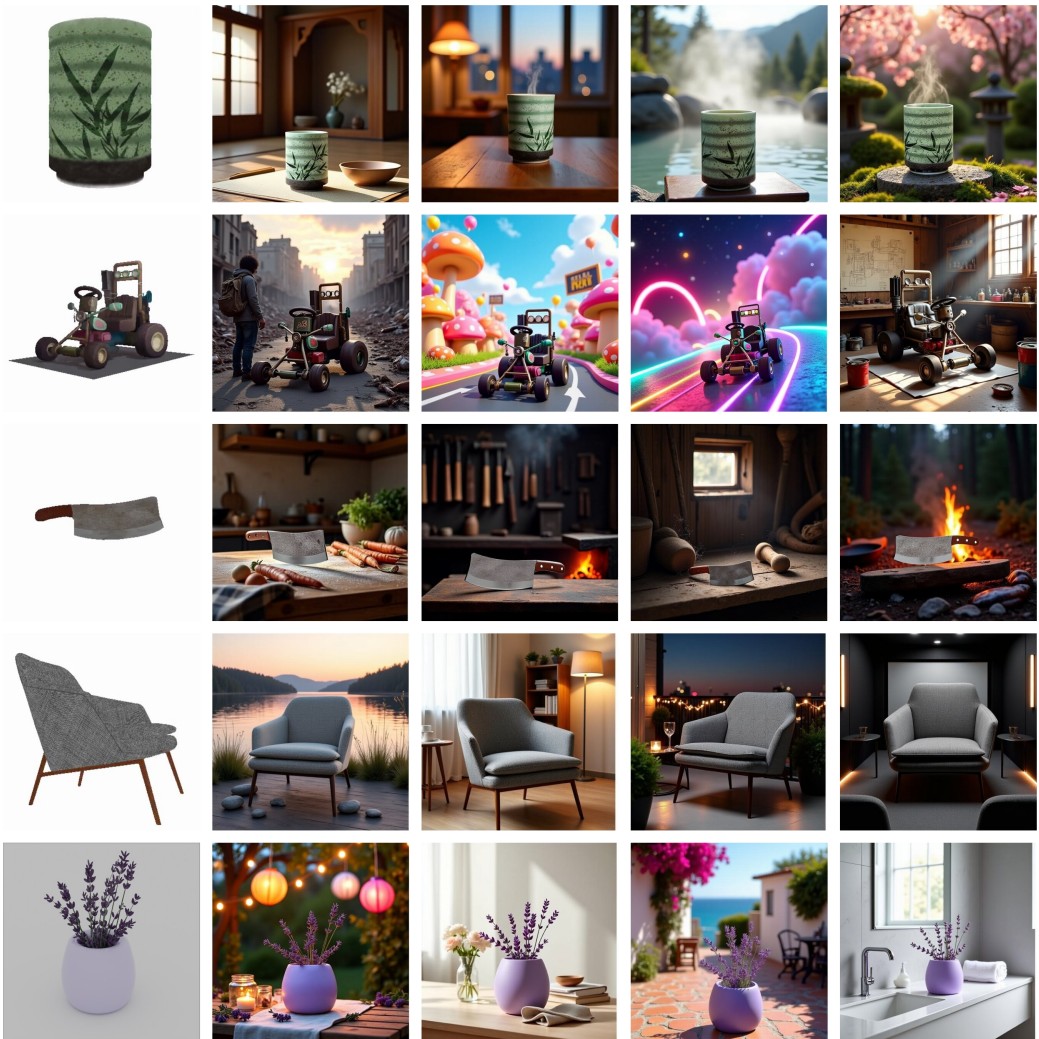

Figure 12: Qualitative results with different 3D assets as references.

