# OpenReview forum: "RefAny3D: 3D Asset-Referenced Diffusion Models for Image Generation"
_ICLR.cc/2026/Conference — ICLR 2026 Poster_

### Official Review · Reviewer_tm5m · 2025-10-28

**Soundness:** 3
**Presentation:** 4
**Contribution:** 3
**Rating:** 6
**Confidence:** 3

**Summary:**

This paper presents RefAny3D, a diffusion-based framework for image generation conditioned directly on 3D assets, enabling consistent alignment between generated images and 3D geometry.
Using multi-view renderings of the 3D asset along with their point maps as conditioning inputs, RefAny3D introduces a dual-branch diffusion architecture that jointly predicts RGB images and point maps, thereby establishing pixel-level correspondences between texture and geometry.
The model employs Shared Positional Encoding (SPE) for spatial correspondence between rgb image and point map, and combines Domain-specific LoRA with Text-agnostic Attention to decouple texture and geometry domains.
For training, the authors construct a pose-aligned dataset derived from Subjects200k using Grounding DINO, Hunyuan3D and Foundation Pose.
Experimental results demonstrate consistent improvements over existing baselines on CLIP, DINO, GIM metrics, and GPT-based evaluations, particularly in maintaining geometric and textural consistency.

**Strengths:**

1. Novelty lies in using 3D assets as direct conditioning for diffusion models, addressing a key gap in 2D reference-based generation by ensuring geometric consistency.
2. The dual-branch architecture is well-motivated and technically sound . Claims are strongly supported by comprehensive experiments and ablations.
3. The paper is clearly written, well-organized, and easy to follow. The proposed architecture is well-illustrated, and component functions are explicitly defined .
4. This work offers an effective and natural solution to the major challenge of spatial consistency in reference-based generation. The joint modeling of RGB and point maps significantly enhances 3D awareness, representing an important contribution to the field.

**Weaknesses:**

1. Missing "Render-and-Edit" Baseline: The introduction mentions a straightforward "render-and-edit" alternative, noting its foreground-background inconsistency issues . However, this intuitive baseline was not included in the experimental comparisons . Empirically demonstrating RefAny3D's superiority over this method would significantly strengthen the paper's claims.
2. Dataset Transparency and Reproducibility: While the data construction pipeline is detailed , key statistics about the final dataset (e.g., scale, category coverage) and its release status are omitted. Providing these details is crucial for transparency and enabling reproducibility.
3. Lack of Failure Analysis: The paper admits poor performance on non-rigid objects due to dataset limitations  but provides no qualitative examples or analysis. It is unclear if this is purely a data issue or an architectural limitation (e.g., rigid spatial alignment). Including such an analysis would make the work more complete.

**Questions:**

The paper sets the number of reference views to N = 8.
Have the authors tested other values (e.g., 4, 12, 16)?
How sensitive is the performance to the number and distribution of viewpoints?
Understanding this sensitivity would clarify the scalability and generalization of the approach.

---

> ### Author Response · Authors · 2025-11-24
>
> Thank you very much for your review. We carefully address your questions and comments in detail below.
>
> > **W1:** Missing "Render-and-Edit" Baseline: The introduction mentions a straightforward "render-and-edit" alternative, noting its foreground-background inconsistency issues . However, this intuitive baseline was not included in the experimental comparisons . Empirically demonstrating RefAny3D's superiority over this method would significantly strengthen the paper's claims.
>
> **Response:** We appreciate the suggestion. We have added a “Render-and-Edit” baseline using Qwen-Image-Edit-2509, with qualitative comparisons provided in Fig. 8 of the paper. Image-editing models tend to partially copy and paste pixels from the rendered input, which can lead to foreground–background, for example, **realistic backgrounds conflicting with rendered foreground assets or the edited object appearing to float unnaturally** (e.g., the helmet in Fig. 8). Moreover, **this approach only preserves information visible in the rendered viewpoint, and occluded regions are often hallucinated**, often producing geometry or textures that contradict the 3D asset (e.g., the tail of the tank in Fig. 8).  **In contrast, our method models 3D awareness and is not restricted to a single visible viewpoint, enabling consistent generation aligned with the full 3D asset.** Furthermore, our approach can build upon existing editing models, transforming them into models that are aware of the 3D structure of the asset. We have added this discussion to the revised paper.(Line 513)
>
> > **W2:** Dataset Transparency and Reproducibility: While the data construction pipeline is detailed , key statistics about the final dataset (e.g., scale, category coverage) and its release status are omitted. Providing these details is crucial for transparency and enabling reproducibility.
>
> **Response:** We thank the reviewer for the suggestion. Our dataset is derived from Subjects200k, which contains 51 annotated object categories. **Our final curated dataset retains coverage across all 51 categories.** We further apply two quality-control checks to ensure geometric and appearance consistency: (1) a Mask IoU filter (>0.8) between the rendered 3D asset and the image mask to remove pose or localization failures, and (2) an LPIPS filter (<0.3) between the rendered asset and the corresponding image region to remove texture mismatches or low-quality mesh reconstructions. **After this filtering, the remaining dataset contains 43k high-quality samples.** **Regarding release status, upon acceptance we will open-source the full pose-aligned dataset.**
>
> > **W3:** Lack of Failure Analysis: The paper admits poor performance on non-rigid objects due to dataset limitations but provides no qualitative examples or analysis. It is unclear if this is purely a data issue or an architectural limitation (e.g., rigid spatial alignment). Including such an analysis would make the work more complete.
>
> **Response:** We appreciate the reviewer for pointing out the limited analysis of failure cases. We have now added a discussion in Line 535 and Fig. 10 of the revised paper. Specifically, because the 3D models in our dataset correspond exactly to the objects in the images and lack non-rigid deformation, our method inherits this limitation. As a result, for non-rigid assets such as ropes and cushions, the generated images tend to retain rigidity and fail to exhibit physically plausible adaptation to the environment.
>
> > **Q1:** The paper sets the number of reference views to N = 8. Have the authors tested other values (e.g., 4, 12, 16)? How sensitive is the performance to the number and distribution of viewpoints? Understanding this sensitivity would clarify the scalability and generalization of the approach.
>
> **Response:** We thank the reviewer for the suggestion. Due to computational resource constraints, we were unable to run experiments with 12 or 16 views. However, we have conducted additional experiments with 4 and 6 views, and compared them with our original 8-view model. **As shown in Fig.8, our method remains effective even with fewer views, and the performance consistently improves as the number of views increases.** We have added this discussion and the corresponding results to the revised paper.(Line 509)

---

> > ### Comment · Reviewer_tm5m · 2025-11-27
> >
> > Thank you for responding to my questions. I would like to maintain my score.

---

### Official Review · Reviewer_RqVJ · 2025-10-28

**Soundness:** 2
**Presentation:** 3
**Contribution:** 2
**Rating:** 4
**Confidence:** 4

**Summary:**

This paper proposes RefAny3D, a 3D asset-referenced diffusion framework for image generation. The model jointly synthesizes RGB images and point maps through a spatially aligned dual-branch architecture to ensure geometry–texture consistency. It further introduces domain-specific LoRA and text-agnostic attention to decouple visual and structural domains. Experiments show clear improvements over 2D reference-based baselines in geometric fidelity and visual quality.

**Strengths:**

1. The paper is clearly structured and well-written, with smooth logical flow from motivation to methodology and results.
2. The work introduces a novel 3D asset-referenced diffusion framework that bridges the gap between 2D reference-based generation and 3D-aware synthesis.
3. The experimental section is comprehensive, including qualitative, quantitative, and ablation studies that convincingly demonstrate the superiority of the method.

**Weaknesses:**

1. Motivation. The paper employs 3D assets as conditioning inputs to ensure geometry–texture consistency; however, generating 2D images does not inherently require multi-view conditioning. The authors should clarify the motivation for introducing 3D asset-based conditioning in this context.
2. Task Definition. Given that the 3D asset is already available, there exist simpler approaches to achieve similar results—for example, rendering the desired viewpoint as a conditioning image and feeding it into an inpainting or editing model. The authors should justify why their proposed framework is necessary compared to these straightforward alternatives.
3. Controllability. The proposed method lacks explicit control over the camera viewpoint of the generated image, which seems inconsistent with the premise of conditioning on a 3D asset. The authors are encouraged to discuss this limitation and explain how viewpoint control could be incorporated.
4. Missing References. The Subject-Driven Generation section in the related work only covers a group of image-guided generation methods, where several related 3D-guided generation methods are missed, such as:
- Wu R, Liu R, Vondrick C, et al. Sin3dm: Learning a diffusion model from a single 3d textured shape
- Wu R, Zheng C. Learning to generate 3d shapes from a single example
- Wang Z, Wang T, Hancke G, et al. Themestation: Generating theme-aware 3d assets from few exemplars
- Wang Z, Wang T, He Z, et al. Phidias: A generative model for creating 3d content from text, image, and 3d conditions with reference-augmented diffusion

**Questions:**

Please refer to Weaknesses.

---

> ### Author Response · Authors · 2025-11-24
>
> Thank you very much for your review. We carefully address your questions and comments in detail below.
>
> > **W1:** Motivation. The paper employs 3D assets as conditioning inputs to ensure geometry–texture consistency; however, generating 2D images does not inherently require multi-view conditioning. The authors should clarify the motivation for introducing 3D asset-based conditioning in this context.
>
> **Response:** We appreciate valuable feedback regarding the motivation for introducing 3D asset conditioning.
>
> The core motivation for our paper is to seamlessly integrate a specific 3D asset into the generated image using the powerful synthesis capabilities of a Text-to-Image (T2I) model.  Specifically, we aim to **generate diverse images using only text prompts, without requiring manually specify complex rendering parameters** such as the 3D asset’s placement, camera pose, or lighting conditions.
>
> While it is true that general 2D image generation does not inherently require multi-view conditioning, **conditioning only on a single rendered view of the 3D asset inherently limits the model to generating only that one perspective.** **In different scenes, an object should naturally adopt a suitable pose.** For example, an axe embedded in a tree stump while chopping wood has a very different plausible orientation compared to an axe displayed upright in a museum. If we impose a fixed and unsuitable object viewpoint as a conditioning signal, the model may be forced to generate unnatural or unreasonable results that do not match the scene context.
>
> When generating images with 3D assets, we prioritize showcasing diverse viewpoints of the 3D objects, ensuring their seamless integration with their respective backgrounds. Our model receives all necessary geometric and textural information. This allows us to guarantee geometry-texture consistency from any viewpoint.
>
> > **W2:** Task Definition. Given that the 3D asset is already available, there exist simpler approaches to achieve similar results—for example, rendering the desired viewpoint as a conditioning image and feeding it into an inpainting or editing model. The authors should justify why their proposed framework is necessary compared to these straightforward alternatives.
>
> **Response:** We thank the reviewer for the insightful suggestion. To address this point, we implemented the straightforward approach of first rendering the 3D asset into an image and then applying an editing model to modify it. However, this approach requires manual selection of viewpoints, object placement, and careful tuning of the renderer. Poor choices can easily lead to foreground-background mismatches or unrealistic floating objects. **Moreover, this method is inherently limited to the rendered viewpoint; when generating novel views, it often hallucinates content that does not exist in the 3D asset.**
>
> We additionally provide qualitative comparisons in Fig. 8 of the paper. Image-editing models tend to partially copy–paste pixels from the rendered input, which leads to foreground–background inconsistency, for example, realistic backgrounds conflicting with rendered foreground assets and the edited object appearing to float unnaturally (e.g., the soldier helmet in Fig. 8). Moreover, this approach preserves only information visible from the rendered viewpoint, and occluded regions are often hallucinated, resulting in geometry or textures that contradict the 3D asset (e.g., the tank in Fig. 8). In contrast, our method models 3D awareness and is not limited to a single viewpoint, enabling consistent generation aligned with the full 3D asset.

---

> ### Author Response · Authors · 2025-11-24
>
> > **W3:** Controllability. The proposed method lacks explicit control over the camera viewpoint of the generated image, which seems inconsistent with the premise of conditioning on a 3D asset. The authors are encouraged to discuss this limitation and explain how viewpoint control could be incorporated.
>
> **Response:** We appreciate the reviewer’s insightful comment on viewpoint controllability. Our method stems from a different design idea. By leveraging the powerful generative priors of diffusion models, we aim to allow the 3D object to naturally integrate into the 2D image, without the need for extensive manual specifications of camera poses, object placement, and other complex configurations required by traditional rendering pipelines.
>
> In practice, our framework still enables meaningful viewpoint variation. By modifying the prompt or adjusting the 3D asset’s local coordinate system, the generated images exhibit diverse perspectives, as illustrated in Fig. 9. We have included this new figure in the revised paper. This provides flexible and intuitive viewpoint control without requiring explicit camera calibration.
>
> We agree that explicit camera-conditioned control could further broaden the practical applicability of our method. Incorporating camera-guided conditioning represents a promising direction for future research and may further enhance viewpoint controllability within our framework.
>
> > **W4:** Missing References. The Subject-Driven Generation section in the related work only covers a group of image-guided generation methods, where several related 3D-guided generation methods are missed.
>
> **Response:** We thank the reviewer for the suggestion. We have added the missing 3D-guided generation works to the Related Work section

---

### Official Review · Reviewer_CgtG · 2025-10-31

**Soundness:** 2
**Presentation:** 3
**Contribution:** 1
**Rating:** 2
**Confidence:** 4

**Summary:**

This paper studies how to synthesize photorealistic images of a specific 3D object given the 3D object model, a straightforward and trivial problem given the current literature. The 3D object is first rendered to multi-view RGB images and point maps encoding the per-view geometry. Then, the stylized image is generated simply via conditioning on the rendered images and point maps.

**Strengths:**

Product-wise, this task is very useful for commercializing generative models for rendering different photos of specific object products, mostly like a diffusion model shader.

**Weaknesses:**

There is too little technical contribution in this paper. Although the pipeline works, it is a straightforward engineering pipeline. I believe this is very practical for industry and product applications, but far below the standard of an ICLR paper.

**Questions:**

I only have one simple question, why is this method novel?

---

> ### Author Response · Authors · 2025-11-24
>
> We thank the reviewer for recognizing the effectiveness and practical value of our method. Our task aims to embed a 3D asset into a 2D image while preserving precise multi-view consistency, which is technically non-trivial and not fully addressed by existing methods. As shown in Fig. 8, straightforward extensions of current approaches fail to maintain geometric fidelity and lead to incoherent foreground–background compositions. This demonstrates that a simple straightforward engineering solution is insufficient for this demanding problem.
>
> To this end, we introduce four technical contributions to enhance 3D structure awareness:
>
> (1) **Dual-Branch Generation Strategy.** We adopt a dual-branch generation strategy that jointly outputs RGB and point maps, enabling more 3D structure-aware modeling of the asset.
>
> (2) **Shared Positional Embedding.** We employ cross-domain shared positional embeddings to preserve spatial alignment between the RGB and point-map domains.
>
> (3) **Domain-Specific LoRA.** We design domain-specific LoRA modules to effectively disentangle the RGB and point-map domains during generation.
>
> (4) **Text-Agnostic Attention.** We introduce a text-agnostic attention mechanism to address the inherent information asymmetry between point maps and RGB images, particularly in background regions.

---

### Official Review · Reviewer_fJ67 · 2025-11-01

**Soundness:** 3
**Presentation:** 3
**Contribution:** 3
**Rating:** 6
**Confidence:** 3

**Summary:**

The paper introduces RefAny3D, a 3D-asset–referenced diffusion framework that jointly generates an RGB image and its point map conditioned on a rendered multi-view set of a reference 3D asset plus text. Core ideas include: (1) a spatially aligned dual-branch architecture with shared positional encodings so RGB and point-map tokens stay pixel-aligned; (2) Domain-specific LoRA (Domain-LoRA for point-map tokens and Reference-LoRA for appearance/reference tokens) and a text-agnostic attention design to reduce cross-domain interference; and (3) a new pose-aligned dataset constructed by extracting objects with GroundingDINO/SAM, generating meshes with Hunyuan3D, estimating 6D pose with FoundationPose, and rendering point maps aligned to the images. Experiments show improved geometric/texture consistency versus baselines (DreamBooth, IP-Adapter, DSD, OminiControl), with comprehensive ablations and qualitative/quantitative comparisons.

**Strengths:**

1. The pipeline that pairs real images with mesh assets (via Hunyuan3D + FoundationPose) to obtain aligned point maps is practically useful for research on 3D-aware image diffusion.

2. Domain-LoRA and Reference-LoRA help the network simultaneously generate point maps and RGB, improving stability and disentanglement.

3. The paper is easy to follow, with sound motivation and diagrams.

4. Ablations and comparisons are extensive; qualitative results show crisper textures and better geometric adherence than baselines.

**Weaknesses:**

1. Potential supervision noise from image-to-3D. The dataset relies on image-to-3D generators, which may not perfectly preserve reference fidelity, injecting bias into the training signal. The paper should quantify how frequently generator artifacts or pose errors degrade the learned 3D-conditioned diffusion, and propose mitigation.

2. It remains unclear how much of the gain comes from generating point maps versus simply conditioning on multi-view images; a rigorous comparison against a “no-point-map” generator (or a latent geometric proxy) would clarify necessity.

3. No user study. Given the goal (faithful, identity-preserving generation consistent with a 3D asset), a human preference study would strengthen claims about perceptual quality and faithfulness.

**Questions:**

More details of the dataset construction pipeline should be provided, such as data filtering and robust checking.

---

> ### Author Response · Authors · 2025-11-24
>
> Thank you very much for your review. We carefully address your questions and comments in detail below
>
> > **W1:** Potential supervision noise from image-to-3D. The dataset relies on image-to-3D generators, which may not perfectly preserve reference fidelity, injecting bias into the training signal. The paper should quantify how frequently generator artifacts or pose errors degrade the learned 3D-conditioned diffusion, and propose mitigation.
>
> **Response:** We thank the reviewer for raising this important point. We agree that imperfect reconstructions could introduce noise into the 3D-conditioned diffusion training. In practice, we already adopt a strict quality-based filtering mechanism to ensure high-fidelity supervision. Specifically, for each image–mesh pair, we render the reconstruceted mesh according to the object's pose and compute IoU and LPIPS between the rendered views and the original object image. Only samples whose metrics exceed predefined thresholds (IoU > 0.8, LPIPS < 0.3) are included in the final training set. This filtering removes the majority of artifacts such as 3D generation failure, or incorrect poses. We have added the details of this data filtering procedure in the paper (Line 291).
>
>
> > **W2:** It remains unclear how much of the gain comes from generating point maps versus simply conditioning on multi-view images; a rigorous comparison against a “no-point-map” generator (or a latent geometric proxy) would clarify necessity.
>
> **Response:** We appreciate the reviewer for highlighting this important question about the necessity of the point-map branch. We conducted an additional no-point-map baseline in the paper, where the diffusion model is conditioned solely on multi-view images without any modules for geometric modeling. As shown in Fig.8 of the paper, **without any pointmap modeling, the model struggles to establish correspondences across different viewpoints**, leading it to reference incorrect image regions and produce erroneous results.
>
> > **W3:** No user study. Given the goal (faithful, identity-preserving generation consistent with a 3D asset), a human preference study would strengthen claims about perceptual quality and faithfulness.
>
> **Response:** We thank the reviewer for the suggestion. We have conducted a user study, as shown in the table below, to evaluate generation quality along Faithfulness, Identity, Aesthetic Quality, and Overall Rank.
>
> (1) Faithfulness measures the strict consistency with the input 3D model, ensuring the generated image accurately reflects the geometry and details of the 3D asset.
>
> (2) ID (Identity) measures the consistency of the object's core identity.
>
> (3) Aesthetic Quality evaluates the overall artistic and visual appeal of the generated image.
>
> (4) The Rank metric represents the overall preference ranking of the generated images from different methods under the same 3D object reference, where a lower rank is better.
>
> For (1)(2)(3) metrics, users were asked to rate each generated image on a 5-point scale, ranging from 1 (not at all relevant) and 5 (perfectly relevant). As demonstrated by the results, our method achieves the highest scores in both Faithfulness and Identity, which directly addresses the core goal of "faithful, identity-preserving generation." Furthermore, we also achieve a competitive Aesthetic Quality score and the best overall Rank. This user study has already been incorporated into the paper (Line 405).
>
> | Method | Faithful↑ | ID↑ | Aesthetic↑ | Rank↓ |
> | --- | --- | --- | --- | --- |
> | Textual Inversion | 2.182 | 3.053 | 3.526 | 5.158 |
> | DreamBooth | 3.836 | 4.315 | 4.421 | 2.526 |
> | IP-Adapter | 1.982 | 2.947 | 2.053 | 5.737 |
> | DSD | 4.145 | 4.368 | 4.158 | 2.842 |
> | OmniControl | 3.909 | 4.263 | 4.526 | 3.158 |
> | **Ours** | **4.655** | **4.737** | **4.632** | **1.579** |
>
> > **Q1:** More details of the dataset construction pipeline should be provided, such as data filtering and robust checking.
>
> **Response:** We thank the reviewer for the suggestion. We have updated Section 3.3 (Datasets) to include detailed descriptions of our data filtering pipeline. Specifically, to ensure reliable pose estimation, we calculate the Mask IoU between the object’s 2D mask in the image and the mask rendered from the pose-aligned 3D model, retaining only those samples with an IoU > 0.8. This filtering method excludes failure cases such as incorrect orientations, mismatched viewpoints, or object localization errors. To further ensure that the generated 3D mesh truly matches the appearance of the object in the 2D image, we place the 3D asset at the estimated pose and compute the LPIPS between the rendered object and the corresponding image region, keeping only samples with a LPIPS < 0.3. This step filters out texture mismatches or reconstruction artifacts in the 3D assets.These two complementary checks, Mask IoU for geometric and pose alignment and LPIPS for texture and appearance fidelity, form a reliable data filtering pipeline.

---

### Meta-Review · Area_Chair_xSoB · 2026-01-11

**Summary:**

Scores were initially mixed (6,6,4,2), with reviewers commenting on the lack of a user study, lack of technical novelty, unclear motivations, and missing references and comparisons to baselines. Author rebuttals seem to have addressed all these concerns, with the

**Reviewer Concerns:**

Many concerns were raised, and all but one seem to have been addressed. A full list is provided here:

- Lack of human preference user study
- Missing details about dataset construction and filtering
- Motivation for 3D asset conditioning
- Comparison to simpler "Render-and-Edit" baselines
- Lack of explicit camera viewpoint control
- Missing references for 3D-guided generation
- Request for dataset statistics and release
- Lack of failure analysis for non-rigid objects
- Performance sensitivity to number of reference views

The only outstanding issue seems to be the Reviewer CgtG's complaints about a lack of technical novelty.

**Reviewer Scores:**

Scores were initially mixed (6,6,4,2), and I expect the author rebuttal may have swayed the borderline-negative review to increase their score to a 5.

I expect scores would have shifted to (6,6,5,2). The negative review received not much of a response, but was also an unusually short review.

---

### Decision · Program_Chairs · 2026-01-26

Accept (Poster)